# In Silico Evaluation of Potential NDM-1 Inhibitors: An Integrated Docking and Molecular Dynamics Approach

**DOI:** 10.3390/ph17121715

**Published:** 2024-12-19

**Authors:** Eduvan Valencia, Mauricio Galvis, Jorge Nisperuza, Vladimir Ballesteros, Fredy Mesa

**Affiliations:** 1Departamento de Química, Universidad Nacional de Colombia, Bogotá 111321, Colombia; edvalenciac@unal.edu.co; 2Facultad de Ingeniería y Ciencias Básicas, Fundación Universitaria Los Libertadores, Cra. 16 No. 63a-68, Bogotá 111221, Colombia; mauricio.galvisp@udea.edu.co (M.G.); jorge.nisperuza@libertadores.edu.co (J.N.); vladimir.ballesteros@libertadores.edu.co (V.B.); 3Grupo de Magnetismo y Simulación G+, Universidad de Antioquia, A.A. 1226, Medellín 050010, Colombia

**Keywords:** NDM-1, molecular docking, molecular dynamics

## Abstract

**Background/Objectives:** Non-fermenting Gram-negative bacteria are resistant to most antibiotics, due to the production of enzymes such as NDM-1. Faced with this challenge, computational methods have become essential for the design of NDM-1 carbapenemase inhibitors, optimizing both the time and cost of the development of new lead molecules. **Methods:** In this study, molecular docking and molecular dynamics (MD) simulations were performed in order to identify effective inhibitors against the NDM-1 enzyme. Protein preparation was carried out using UCSF Chimera and AutoDockTools 1.5.7, while ligands were prepared with MarvinSketch, Avogadro, and AutoDockTools 1.5.7. Molecular docking was run with AutoDock4 and AutoDock4Zn, determining that molecules M26 (−13.23 kcal/mol with AutoDock4 and −13.11 kcal/mol with AutoDockZn) and M25 (−10.61 kcal/mol with AutoDock4 and −11.18 kcal/mol with AutoDockZn) presented the best binding energy affinities with NDM-1. The M26 molecule formed six hydrogen bonds with the enzyme. **Results:** MD simulations, performed with GROMACS, indicated that the NDM-1-M26, NDM-1-M35, and NDM-1-M37 complexes showed conformational stability and flexibility. **Conclusions:** These results suggest that the M26, M37, and M35 ligands have significant potential as leading candidates in the development of new NDM-1 inhibitors, outperforming the antibiotic Meropenem in some respects.

## 1. Introduction

In recent years, there has been growing concern in public health systems worldwide regarding the treatment of infectious diseases, especially those caused by bacteria that are multidrug-resistant to commonly used antibiotics, such as β-lactams. These antibiotics have been widely used for decades, due to their high efficacy, affordability, and low toxicity [1]. However, increasing bacterial resistance poses a significant global pandemic risk, threatening to collapse the healthcare systems of various countries [2]. This situation is largely due to the indiscriminate use of antibiotics, both by self-medication in humans and for animals, which has facilitated the emergence of resistance mechanisms in microorganisms, driven by evolutionary mutations. One of the most common mechanisms is the production of β-lactamases, enzymes that hydrolyze the β-lactam ring of conventional antibiotics, nullifying their bactericidal or bacteriostatic action [3,4,5,6,7]. Among the most resistant bacteria due to carbapenemase production are the non-fermenting Gram-negative bacteria, such as Acinetobacter and Pseudomonas, as well as members of the Enterobacteriaceae family, such as *Klebsiella* spp, *Serratia*, *Proteus*, *E. coli*, and *Enterobacter* spp. In February 2017, the WHO included these pathogens in an urgent priority list for the development of new antibiotics [8]. The rapid spread and high resistance of these pathogens in hospital settings poses a serious public health problem globally, being responsible for approximately 1.27 million deaths in 2019 from infections associated with antimicrobial multidrug resistance [9]. When patients present infections caused by these resistant bacteria, broad-spectrum antibiotics such as carbapenemics, considered the last line of defense, are resorted to [10]. These drugs are more stable against β-lactamases and have a broader spectrum of action than other β-lactams [11]. However, their effectiveness has been compromised by the emergence of carbapenemases such as NDM (New Delhi Metallobetalactamase), which inactivate these drugs, blocking their therapeutic function [12]. Faced with this challenge, research in microbiology and medicinal chemistry has focused on developing new inhibitors without a β-lactam ring, capable of evading bacterial resistance mechanisms, and with better oral bioavailability and toxicity profiles [2,13]. The NDM-1 enzyme, in particular, is composed of a 270-residue chain and a 28-amino acid signal peptide at its N-terminal end, which allows its transport to the periplasmic space and, in some cases, to the outer membrane of Gram-negative bacteria, to hydrolyze β-lactam antibiotics [11]. The active site of this metalloprotein consists of two Zn^2+^ ions (Figure 1), surrounded by the amino acids His-120, His-122, Asp-124, His-189, Cys-208, and His-250. One of the zinc ions exhibits tetrahedral coordination with residues His-120, His-122, His-189, and a hydroxyl group of Asp-124 [14,15]. The second Zn^2+^ cation, on the other hand, forms a trigonal pyramidal coordination with the hydroxyl of Asp-124 and residues Cys-208 and His-250.

The two zinc ions interact with each other at a distance of 3.2 Å, being influenced by the amino acid Asp-124. One of the Zn^2+^, located at the so-called “cysteine site”, exhibits interactions with the oxygen of a solvent molecule at a distance of 2 Å, suggesting the formation of a bond with a hydroxide ion. This hydroxide ion can act as a nucleophile in attacking the carbonyl carbon of the β-lactam ring [15]. However, elucidation of the precise mechanism of action of NDM-1 still requires further investigation [16]. Computer-aided drug design remains one of the most common strategies for the discovery and development of new drugs, allowing cost reduction and shortening research times [17]. Through in silico assays, it is possible to predict values or make estimates of descriptors and properties of interest through probabilistic approximations, using models based on published data and molecular structure analysis. These data provide a rational view of the bioavailability and biosafety profile of compounds, discarding those with undesirable properties. However, judgment is required in screening, as some molecules that appear promising based on these parameters may be orally inactive, while others may be active despite deviating from certain optimal physicochemical ranges [18]. Computational approaches in the design of potential inhibitors of metallo-β-lactamases, such as NDM-1, can increase the probability of success and reduce the costs associated with the development of new lead molecules, as previously mentioned [18]. In the present investigation, an evaluation was carried out using in silico assays of a set of molecules selected from a previous study, identifying 22 compounds with the best oral bioavailability values and low toxicity [19]. These included ethylenediamine derivatives, N,N′,N′′-triacetate-1,4,7-triazacyclonane, phosphonic acid ester mercaptos, sulfur-containing carboxylic acids, dipicolinic acid, cyclic borates, chromones, natural compounds, and thioamide derivatives. The potential inhibitors were subjected to molecular docking calculations against the NDM-1 enzyme (PDB code: 5ZGZ), using the AutoDock4 and AutoDock4Zn programs. The compounds that showed the best binding energy affinities were selected to perform molecular dynamics (MD) simulations for 10 ns, in order to evaluate the stability of the complexes formed with NDM-1. As a result, compounds M26, M35, and M37 were identified as potential inhibitors of the drug target under study.

## 2. Results

### 2.1. Molecular Coupling

A total of twenty-two compounds previously described in the literature were coupled to the crystallographic structure of the NDM-1 enzyme (PDB: 5ZGZ), together with two reference antibiotics (meropenem and imipenem). Of these, four compounds (M25, M26, M35, and M37) stood out as having the best binding affinities in the analyses performed with both AutoDock4 (Table 1) and AutoDock4Zn (Table 2). The binding energies of the selected compounds, as well as the key molecular interactions, including the nature of the bonds and the relevant distances, are documented in the aforementioned tables. These results underscore the potential of the selected compounds to interact efficiently with the active site of NDM-1, positioning them as promising inhibitors in the development of therapeutics against this enzyme.

### 2.2. Intermolecular Contacts

As shown in Table 1 and Table 2, potential hydrogen bonds and hydrophobic interactions between the ligands and the NDM-1 enzyme were analyzed using two molecular docking programs. AutoDockTools 1.5.7 software was used for structure preparation and parameterization. Subsequently, 2D images were generated with Maestro 14.1 (https://www.schrodinger.com/products/maestro-viewer/, accessed on 23 November 2024) after docking analysis with AutoDock4Zn (Figure 2), a software that employs a specialized force field for metalloenzymes.

The results highlighted key residues of the NDM-1 enzyme involved in interactions with ligands M26, M25, M37, and M35, highlighting their role in stabilizing the complexes. In particular, compounds M26 and M25 showed more than three intermolecular hydrogen bonds with the amino acids, ions, and water molecules of the enzyme active site, indicating a high degree of interaction with their catalytic environment.

This multidimensional approach provided a detailed insight into the possible inhibition mechanisms that could be exerted by the studied molecules on the NDM-1 drug target, underlining their potential as promising candidates for future experimental trials [8].

### 2.3. Molecular Dynamics Trajectories

The structural stability results, simulation convergence, and ligand effects on NDM-1 conformation are reported below.

Figure 3 shows the calculation of the RMSD for all the systems analyzed, which allowed us to evaluate the possible conformational changes (in terms of flexibility of the NDM-1 backbone), as well as the stability, convergence, and effects of the ligands on the protein during the 10 ns simulation. For the four systems under study, it was observed that the oscillations experienced by the complexes during the molecular dynamics remained below 3 Å, which was favorable compared to the NDM-1-Meropenem reference system, in terms of conformational stability.

Structural and dynamic flexibility results of NDM-1 using RMSF data.

Figure 4 shows the RMSF calculations and diagrams for the systems under study, where the mobility and conformational flexibility of the NDM-1 backbone was evaluated in the presence of the different ligands. The plots obtained show the fluctuations experienced by the protein in the different regions of the amino acids that compose it, which were similar to those observed in the NDM-1-Meropenem reference complex.

Results of intermolecular hydrogen bonding interactions of NDM-1 with ligands.

Figure 5 evidences the hydrogen bonding interactions generated between the protein and the possible inhibitors. This suggests that such interactions could have contributed to the stability of the complex throughout the molecular dynamics simulations performed.

## 3. Discussion

### 3.1. Analysis of Molecular Couplings

Of the initial 22 compounds, M26 presented the lowest binding affinities against the NDM-1 protein, with values of −13.23 kcal/mol using AutoDock4 and −13.11 kcal/mol with AutoDockZn, while M25 showed values of −10.61 kcal/mol and −11.18 kcal/mol, respectively. Furthermore, compounds M25, M35, and M37 experienced improvements in their binding affinity values when using AutoDockZn (Table 2) compared to AutoDock4 (Table 1), evidencing significant differences in the results obtained. These discrepancies can be attributed to the advanced features of AutoDockZn, which extends the AutoDock force field by including a specialized potential to describe both the geometric and energetic components of the ligand–protein interaction. In particular, AutoDockZn improved the representation of the coordination interactions between ligands and zinc ions in the active site of NDM-1, allowing a more accurate reproduction of the coordination geometry. These improvements are essential for optimizing specific interactions with zinc ions, a key objective in the design of metalloenzyme inhibitors such as NDM-1 [20].

### 3.2. Analysis of Intermolecular Contacts

All molecules with the best couplings were located in the active site of NDM-1, interacting with amino acids coordinating zinc atoms. Molecule M26 stands out for forming four hydrogen bonds, including those with water molecule 303, which participates in the coordination of zinc ions and is involved in the hydrolysis mechanism of NDM-1. In addition, it establishes interactions with Lys 211, which contributes to the structural stability and functionality of the active site, and with residue Asn 220. Among the hydrophobic interactions of M26 are those with Trp 93, Val 73, Cys 208, Ala 121, and Met 67 (Figure 2b). The M25 molecule forms three hydrogen bonds: one with the water molecule, one with Lys 211, and one with Asn 220. Its hydrophobic interactions include residues Met 154, Met 67, Trp 93, and Val 73 (Figure 2a). In the case of M37, hydrogen bonds were observed with residues Lys 211 and Asn 220, while the hydrophobic interactions encompassed Val 73, Trp 93, Met 67, and Ala 74 (Figure 2d). Finally, molecule M35 exhibited two hydrogen bonds with residue Lys 211 and one with Asn 220, along with significant hydrophobic interactions with Trp 93, Val 73, Ala 74, and Met 67 (Figure 2c).

The analyses carried out identified functional groups that play a crucial role in the interaction of the ligands with the active site of NDM-1. The carbonyl and amide groups present in compounds M25, M26, M35, and M37 were decisive in the formation of hydrogen bonds with key catalytic residues such as Lys211 and Asn220. In addition, the interaction with the water molecule 303, which coordinates the zinc ions in the active site, reinforces the stability of the complex. These interactions were particularly noticeable in the case of compound M26, which presented the highest number of hydrogen bonds, standing out as a ligand with high inhibitory potential.

In the 2D diagrams (Figure 2) of the molecules studied, a large number of hydrophobic interactions stand out. Previous research has shown that these interactions, followed by hydrogen bonds, are determinant for the stability of the protein–ligand complex. Furthermore, it has been proposed that a higher number of hydrophobic interactions is advantageous, especially in small ligands, as these interactions favor structural adaptation in the active site environment by overcoming electrostatic geometrical constraints.

In summary, it was possible to determine that the selected compounds presented key interactions in the active site of NDM-1, which explained their better results in the analyses. In particular, M26 showed the lowest coupling energy and formed multiple hydrogen bonds with residues such as Asn220 and Lys211, as well as with the water molecule 303, which coordinates the Zn^2+^ ions essential for the catalytic activity of the enzyme. These interactions increased the conformational stability of the complex, which was reflected in consistent RMSD values and in the ability of the compound to maintain a sustained interaction in the molecular dynamics simulations, as shown below. Additionally, hydrophobic interactions with nearby residues contributed to the structural adaptation in the active site environment, favoring the coupling and reinforcing the affinity of these compounds. This balance between specific and hydrophobic interactions highlights their potential as effective inhibitors, standing out from the other compounds evaluated. It is important to highlight that while molecular docking studies are fundamental tools for predicting initial interactions between ligands and proteins, their utility can be limited when analyzing dynamic biological systems. Static molecular docking predictions do not always reflect the stability of complexes under more realistic conditions, such as those found in a cellular environment, where factors such as receptor flexibility, solvation, and intermolecular interactions play a crucial role. For this reason, in the present study, molecular dynamics calculations were performed that allowed overcoming some of these limitations, evaluating the conformational stability and flexibility of NDM-1-ligand complexes under dynamic and solvated conditions. Nevertheless, it is essential to perform in vitro biological assays to validate the results obtained through the in silico approaches used in this research.

### 3.3. Analysis of Molecular Dynamics Trajectories

Analysis of structural stability, simulation convergence, and ligand effects on NDM-1 conformation.

The RMSD (root mean square deviation) analysis provided a comprehensive understanding of the stability and convergence of NDM-1–ligand complexes. The results demonstrated how each ligand influenced the protein’s dynamics and interactions within the active site. NDM-1-M25 SYSTEM (Figure 3a): The RMSD of the protein backbone fluctuated between 0.05 nm and 0.17 nm, remaining below the recommended threshold of 0.3 nm, which reflects the overall structural stability. However, a sharp increase in RMSD during the first 2 ns indicates an initial stabilization phase of the metalloenzyme. The ligand M25 exhibited limited stability, establishing interactions predominantly between 0–2 ns and 6.8–8.5 ns. After 8.5 ns, the ligand’s RMSD increased significantly, reaching 0.25 nm, suggesting a loss of stable binding and a potential detachment from the active site. This behavior highlighted M25’s limited capacity to maintain consistent interactions throughout the simulation, which may undermine its inhibitory potential. NDM-1-M26 SYSTEM (Figure 3b): In this system, both the protein and ligand RMSDs initially fluctuated between 0.05 nm and 0.25 nm during the first 2 ns. Beyond this point, the system stabilized, showing steady-state dynamics with RMSD values consistently ranging from 0.1 nm to 0.22 nm. A peak at 4 ns (0.25 nm) indicated a slight displacement of the ligand from the active site, possibly due to conformational adjustments. Additionally, the peak observed just before 9 ns (0.2 nm) suggests a rotation of the ligand within the binding cavity, facilitating the cleavage and formation of new interactions with active site residues or Zn^2+^ ions. These observations underscored the adaptive and stable interaction profile of M26, making it a strong candidate for NDM-1 inhibition. NDM-1-M35 SYSTEM (Figure 3c): The RMSD fluctuated between 0.1 nm and 0.22 nm, with a notable increase between 5 and 6 ns. This increase likely reflected the enhanced protein flexibility induced by the interaction with M35. While the system remained stable overall, these fluctuations suggest moderate conformational adjustments in response to ligand binding. NDM-1-M37 SYSTEM (Figure 3d): The RMSD ranged from 0.03 nm to 0.22 nm, with noticeable fluctuations around 1 ns and between 5 and 6 ns. During this period, the ligand appeared to drift away from the binding site, correlating with increased protein flexibility. These observations indicate weaker ligand–protein interactions compared to other complexes. COMPARISON WITH REFERENCE DRUG MEROPENEM (Figure 3e): The RMSD of the NDM-1-Meropenem system remained consistently below 0.2 nm, reflecting stable interactions and minimal conformational fluctuations. This stability served as a benchmark for evaluating the studied ligands. Notably, M26 displayed a dynamic behavior similar to Meropenem, suggesting its potential as a comparable inhibitor. The results suggest that, with the exception of the NDM-1-M25 system (based on RMSD data at the end of the 10 ns simulation), the complexes exhibited notable structural stability and relative convergence throughout the simulations. The increases in RMSD observed in some cases (such as in the M26 complex at 4 ns and in M35 between 5 and 6 ns) were consistent with processes of ligand reorientation or adjustment within the active site, a characteristic and expected behavior in molecular dynamics simulations.

Analysis of the structural and dynamic flexibility of NDM-1 using RMSF data.

Analysis of the root mean square fluctuations of NDM-1 protein with different ligands provided detailed information on the most mobile regions and their possible relation to ligand–protein interaction. In the NDM-1-M25 system (Figure 4a), two significant peaks were observed, with values of approximately 0.30 nm and 0.18 nm, which highlighted increased mobility in specific regions of the protein. These fluctuations may have been due to the dynamic interaction of the ligand with key residues, suggesting some structural adaptability in these areas. Comparatively, the NDM-1-Meropenem reference system (Figure 4e) exhibited peaks in similar regions, supporting the importance of these areas for ligand binding and stabilization. In the NDM-1-M26 (Figure 4b) and NDM-1-M37 (Figure 4d) systems, the RMSF values showed similar patterns to NDM-1-M25, but with a smaller fluctuation in the N- and C-terminal regions of the protein. This may reflect a more stable binding of these ligands, which could limit movements in these peripheral regions. The RMSF data for all complexes highlighted two regions of significant fluctuation in the range of residues 62 to 75 and residues 212 to 230, areas that may be involved in ligand stabilization and binding to the active site of NDM-1. However, it is noteworthy that the NDM-1-M35 system (Figure 4c) did not show a prominent fluctuation in the second region (residues 212 to 230), suggesting that this ligand may have had a less stabilizing impact in this area compared to the other complexes studied. These results indicate that the structural flexibility of NDM-1 varies significantly as a function of the ligand bound, with the NDM-1-M26 and NDM-1-M37 systems standing out as the most stable in terms of reduced fluctuations. This may be related to the higher affinity and more efficient interaction with the active site, which reinforces their potential as promising inhibitors of the enzyme.

Analysis of intermolecular hydrogen bonding interactions of NDM-1 with ligands.

Analysis of hydrogen bond dynamics revealed significant differences in the interaction patterns and stability of the complexes studied. In the case of the NDM-1-M25 system (Figure 5a), up to three hydrogen bonds were formed intermittently during the molecular dynamics (MD) simulation. However, periods were identified in which these interactions were lost, which could compromise the overall stability of the complex. This behavior suggests that, although M25 could establish initial interactions, its ability to maintain stable hydrogen bonds over time was limited, which could reduce its effectiveness as an inhibitor. On the other hand, the NDM-1-M26 complex (Figure 5b) stood out as showing the highest number of hydrogen bonds among all systems tested, including the reference drug Meropenem (Figure 5e). During the first 3 ns of the simulation, M26 formed between six and seven hydrogen bonds, stabilizing around three consistent hydrogen bonds from the fourth nanosecond onward. This stability may be attributed to a structural rearrangement of the ligand within the protein binding cavity, which optimizes its interactions with the active site residues. This pattern reinforces M26 as a strong candidate for NDM-1 inhibition. In the case of the NDM-1-M35 system (Figure 5c), one- to two-hydrogen bond interactions were observed during the simulation. However, distortions in the interaction profile at certain time intervals suggested a lower stability compared to M26, possibly due to a suboptimal binding geometry or lower complementarity with the active site residues. For the NDM-1-M37 complex (Figure 5d), hydrogen bonds were maintained until approximately the first 8 ns of the simulation. After this period, the interactions between the metalloenzyme and the ligand decreased significantly, indicating a loss of binding stability. This behavior could be attributed to conformational changes in the ligand or to the dynamic flexibility of the protein–ligand complex. Taken together, the results highlight the superior performance of M26, which not only formed the highest number of hydrogen bonds, but also maintained its stability during the MD simulation. This behavior contrasts with the less consistent interactions observed in M25, M35, and M37, reinforcing the potential of M26 as a prime candidate for NDM-1 inhibition. The ability of M26 to maintain stable hydrogen bonds over time correlated positively with its coupling energies, suggesting a relationship between the docking results and dynamic interaction profiles in this system.

Finally, comparative analysis between the binding energy values obtained in the molecular docking and molecular dynamics simulations (RMSD, RMSF, and H-BOND) suggests a positive correlation for compounds M26, M35, and M37. These ligands, which presented the lowest binding energies in AutoDock4 and AutoDockZn (up to −13.23 kcal/mol for M26), also showed remarkable conformational stability in the 10 ns simulations, with RMSD values of the protein–ligand complex below 3 Å and constant fluxes in hydrogen interactions. M26, in particular, maintained between three and seven stable hydrogen bonds throughout the simulation (Figure 5b), supporting its robust ability to interact in the active site. In contrast, ligands with less favorable binding energies, e.g., M37, showed more pronounced fluctuations in RMSD and temporal losses of critical interactions, indicating a lower overall affinity. This behavior underlines that a low binding energy in molecular docking can predict stable and specific interactions during MD simulations, especially in systems involving metalloproteins such as NDM-1.

## 4. Materials and Methods

Initially, the structure of the metalloenzyme NDM-1 (PDB ID: 5ZGZ) was downloaded from the Protein Data Bank (RCSB, www.rcsb.org) and selected based on parameters such as resolution and free R value, among others. The ligands were selected from a previous investigation carried out by the authors, from which 22 potential inhibitors were chosen. These compounds showed the best values in terms of absorption, distribution, metabolism, excretion, and toxicity (ADMET).

### 4.1. Protein Preparation

The NDM-1 enzyme structure was selected from the RCSB PDB among more than 40 possible proteins, prioritizing several criteria, such as the lowest resolution (5ZGZ = 0.95 Å), the minimum free R value (5ZGZ = 0.135), the number of chains (only chain A, since it acts as a monomer), the percentile rank, and the chain length (242 amino acids). In addition, a structure without mutations was chosen [21]. The metalloenzyme was refined using UCSF Chimera 1.17.3 software (Pettersen et al., 2004) [22] removing water molecules and the EDO ligand (1,2-ethanediol). Missing residues in the crystal structure (28 amino acids) were not included in this study. Subsequently, AutoDockTools 1.5.6 software (Morris et al., 2009) [23] was used to add polar hydrogens and Kollman charges (specific for amino acids). Finally, the protein was saved in PDBQT format for subsequent molecular docking.

### 4.2. Ligand Preparation

The 2D structures of the potential NDM-1 inhibitors (Figure 6) were generated using MarvinSketch 23.12 software (Csizmadia, P., 1999) [24], where the protonation states at pH 7.5 were calculated and saved in mol2 format. The 3D structures, together with their subsequent optimization, were created using Avogadro 1.2.0 software (Hanwell et al., 2012) [25], adding the hydrogens at pH 7.5. Subsequently, energy minimization was carried out using the UFF force fields for the boron- and potassium-containing molecules, and MMFF94 for the other compounds, including the reference antibiotics [22]. The structures were saved in mol2 and PDB format. The next step was to process the ligands with AutoDockTools 1.5.7, where all hydrogens were added, Gasteiger charges were calculated, and non-polar hydrogens were applied. Finally, the molecules were saved in PDBQT format for use in molecular docking [8].

### 4.3. Molecular Coupling

Molecular docking of the evaluated compounds was carried out with the aim of determining the binding energy (ΔG) against the selected drug target. For this purpose, AutoDock software was used, a tool widely recognized for its ability to predict how small molecules, such as substrates or potential drugs, interact with a receptor of known three-dimensional structure [26]. Calculations were performed with AutoDock4 and AutoDock4Zn, assisted by the AMDock 1.6.2 program [27], to allow a comprehensive comparison between the two approaches. AutoDock4Zn was selected due to its ability to incorporate a specialized force field specifically designed for metalloproteins containing zinc atoms in their active structure, as is the case of the enzyme studied [20].

From the preparation of the enzyme and ligands, the size of the GRID BOX was established as a function of the amino acids of the active site of the protein [14] and from the size of the drugs and potential inhibitors. The defined values were as follows: center at X=−4.2 Å, Y=−5.4 Å, Z=−16.2 Å, and size of X=60 Å, Y=60 Å, Z=60 Å.

Subsequently, the programs were configured to perform a maximum number of energetic evaluations of 10,000,000, with a level of completeness equal to 22, which guaranteed an exhaustive exploration of the ligand conformational space. The Lamarckian genetic algorithm, which combined local and global optimization methods to improve the accuracy of the results, was used to generate the output files. Once the couplings had been executed, the conformations generated by the compounds and antibiotics were evaluated in terms of their binding free energy (ΔG) values. The conformations with the most favorable values were selected as the best for further molecular dynamics calculations, allowing a detailed analysis of their stability and behavior in the context of the active site of the drug target.

### 4.4. Molecular Imteractions

The 3D visualization of hydrophobic interactions and hydrogen bonds between ligands and the NDM-1 enzyme was carried out using AutoDockTools 1.5.7 and PyMOL programs [28], which allowed the key interactions in the active site of the protein to be analyzed and graphically represented. To generate the 2D docking images, Maestro Viewer software [29], known for its ability to produce clear and detailed representations of ligand–receptor interactions, was used. This comprehensive approach provided an accurate view of the possible inhibition mechanisms that could be exerted by the studied molecules on the NDM-1 drug target, highlighting their potential as promising candidates for future experimental assays.

### 4.5. Molecular Dynamics

Taking into account the binding free energy values obtained in the molecular couplings of the analyzed compounds, the most promising confomers were selected to perform molecular dynamics (MD) simulations, in order to evaluate the stability and flexibility of the complex during the course of the simulation. For this purpose, the GROMACS 2023.2 software was used [27]. The simulation time was 10 ns in an aqueous environment, using the CHARMM36-Jul2022.ff force field. The solvation box of the protein–ligand complex was dodecahedral in shape, containing water according to the SPC model, and sodium and chloride ions were added to equilibrate the system.

### 4.6. Molecular Dynamics Trajectory

After completion of the molecular dynamics (MD) calculations, periodic boundary condition (PBC) correction was applied. To verify the structural stability and flexibility of the complex, the root mean square deviation (RMSD) and root mean square fluctuation (RMSF), respectively, were calculated using GROMACS scripts. Finally, the number of hydrogen bonds between NDM-1 and ligands along the trajectory was determined using the gmx hbond script. Visualization of the plots was performed using XmGrace [8].

## 5. Conclusions

From a total of 22 initial compounds, M25, M26, M35, and M37 stood out as having the best values in the molecular docking analysis. This study highlighted the advantages of using AutoDockZn compared to AutoDock4 to analyze protein–ligand interactions in systems containing metal ions such as NDM-1. The extended force field included in AutoDockZn not only generates variations in docking results compared to AutoDock4, but also allows obtaining lower binding energy values in some cases, indicating a better affinity of the ligands towards the protein. Furthermore, AutoDockZn improves the representation of metal–ligand interactions, especially the coordination between ligands and zinc ions in the active site. This capability is essential for more accurately modeling these interactions, and is crucial for the rational design of inhibitors targeting metalloenzymes such as NDM-1. The analysis of the intermolecular contacts of M25 and M26 highlighted their interaction with the water molecule 303, located in the active site of NDM-1, which is crucial for the catalytic activity of the enzyme by coordinating with the zinc atoms. Furthermore, hydrophobic, polar, and negatively charged interactions included key active site residues such as Cys 208, His 250, His 122, His 189, and Asp 124. Other amino acids such as Lys 211, Asn 220, Trp 93, Val 73, and Met 67 were also found to be relevant for ligand stabilization in the active site. Root mean square deviation (RMSD) trajectory analysis suggested the formation of complexes with structural stability and relative convergence during dynamic simulations. Likewise, root mean square fluctuation (RMSF) analysis revealed fluctuations in two specific regions of NDM-1, which might play an important role in ligand stabilization and binding. Taken together, the results of molecular docking, intermolecular contacts, and molecular dynamics suggest that ligands M26, M37, and M35, compared to the reference antibiotic Meropenem, could be considered as lead molecules for the development of novel NDM-1 inhibitors.

## Figures and Tables

**Figure 1 pharmaceuticals-17-01715-f001:**
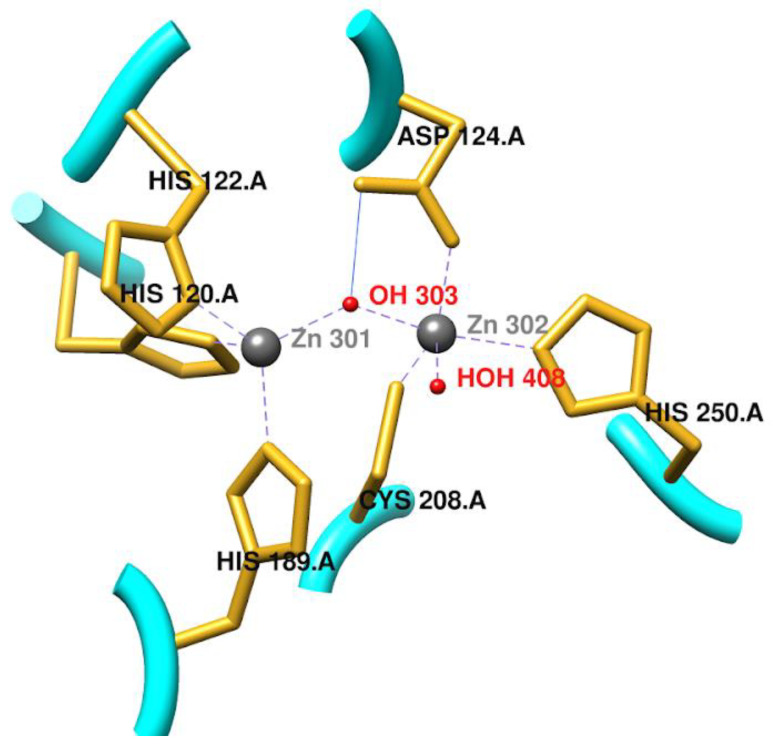
View of the active site of the NDM-1 enzyme. Source—own elaboration.

**Figure 2 pharmaceuticals-17-01715-f002:**
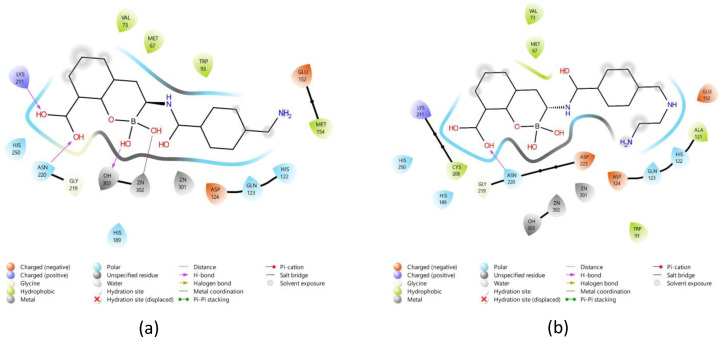
Diagrams showing interactions between NDM-1 residues and molecules. (**a**) M25, (**b**) M26, (**c**) M37, (**d**) M35, and (**e**) Meropenem. Source—own elaboration.

**Figure 3 pharmaceuticals-17-01715-f003:**
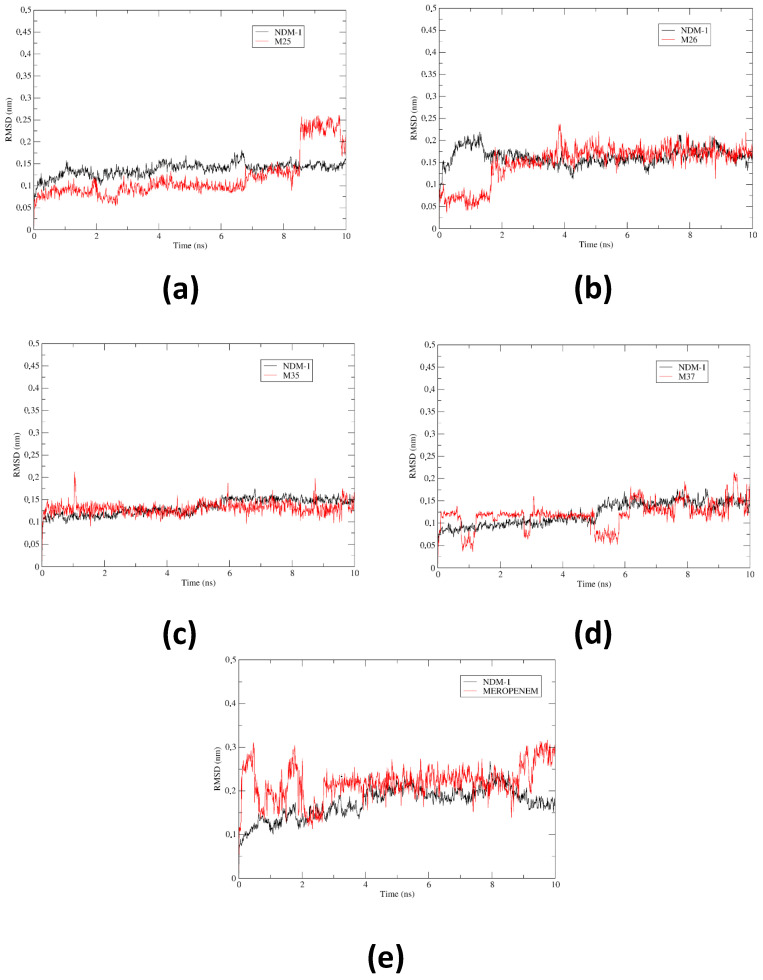
Time evolution of RMSD, during 10 ns MD simulations for (**a**) M25, (**b**) M26, (**c**) M35, (**d**) M37, and (**e**) Meropenem ligands, against the NDM-1 Backbone. Source—own elaboration.

**Figure 4 pharmaceuticals-17-01715-f004:**
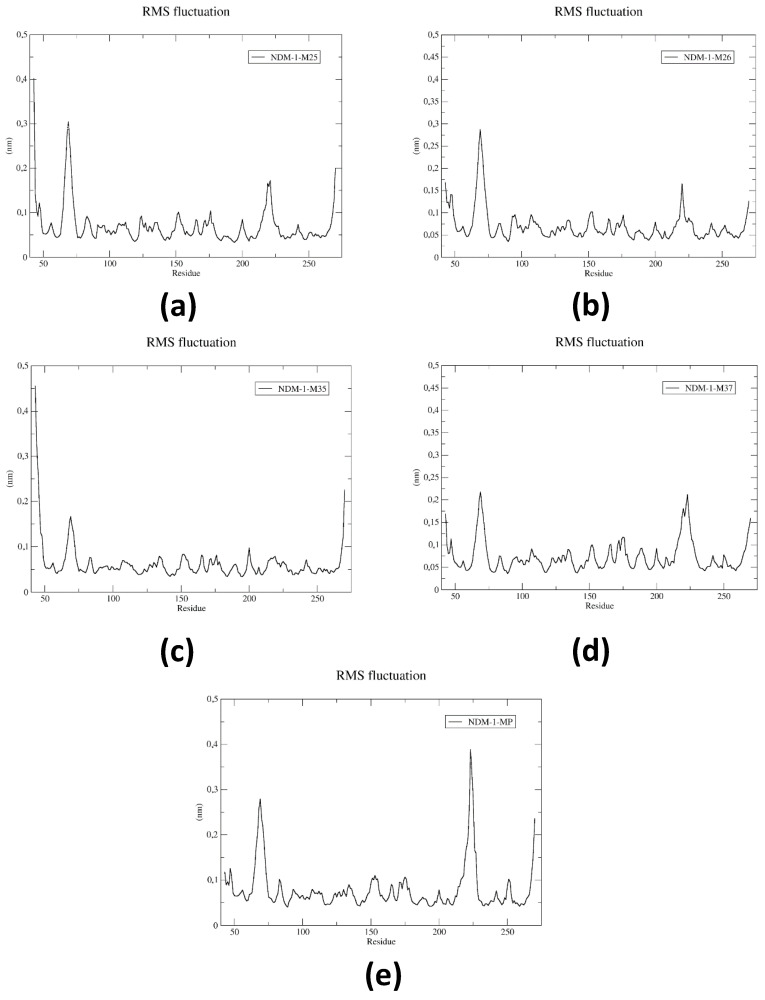
RMSF root mean square fluctuation plot for a 10 ns period of DM simulations for the protein residues of (**a**) NDM-1-M25, (**b**) NDM-1-M26, (**c**) NDM-1-M35, (**d**) NDM-1-M37, and (**e**) NDM-1-Meropenem. Source—own elaboration.

**Figure 5 pharmaceuticals-17-01715-f005:**
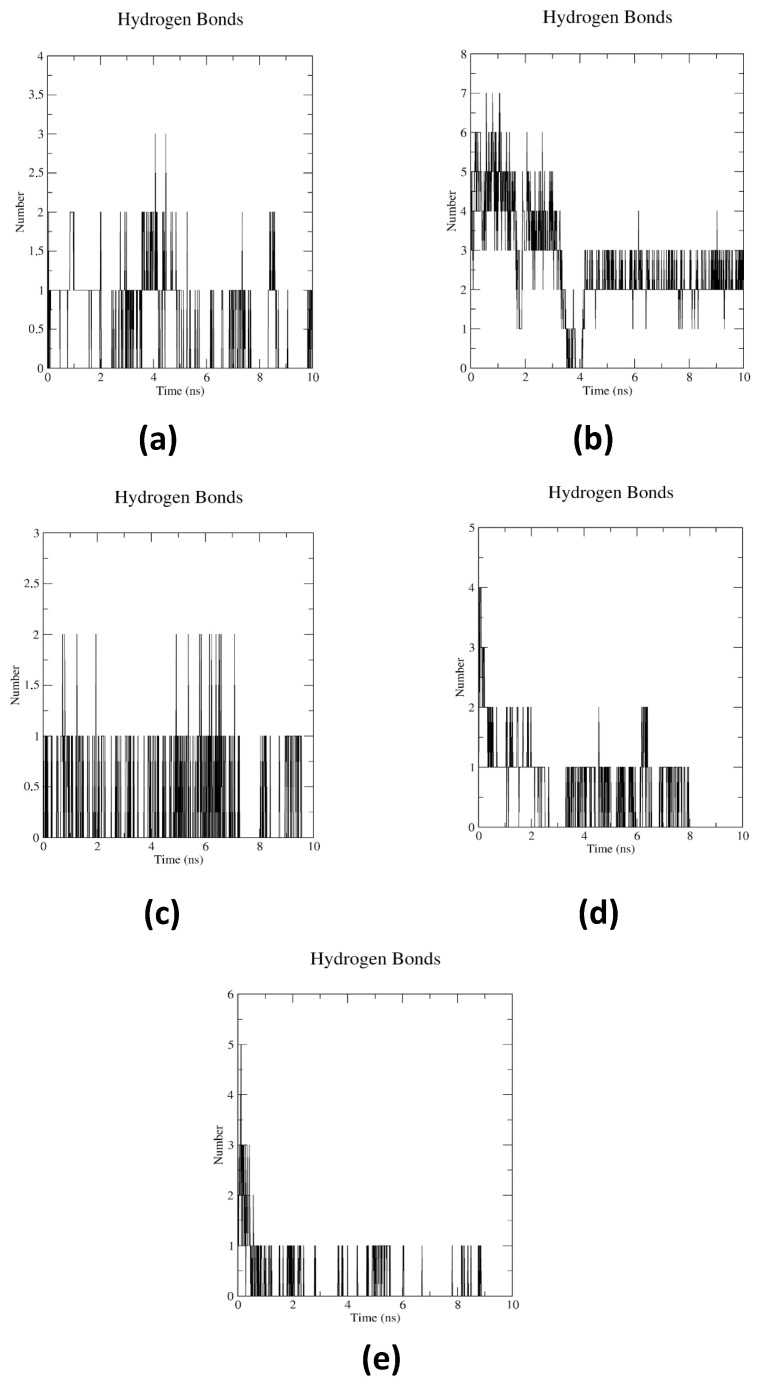
Intermolecular hydrogen bond dynamics for a 10 ns simulation period, for ligands (**a**) M25, (**b**) M26, (**c**) M35, (**d**) M37, and (**e**) Meropenem, against NDM-1. Source—own elaboration.

**Figure 6 pharmaceuticals-17-01715-f006:**
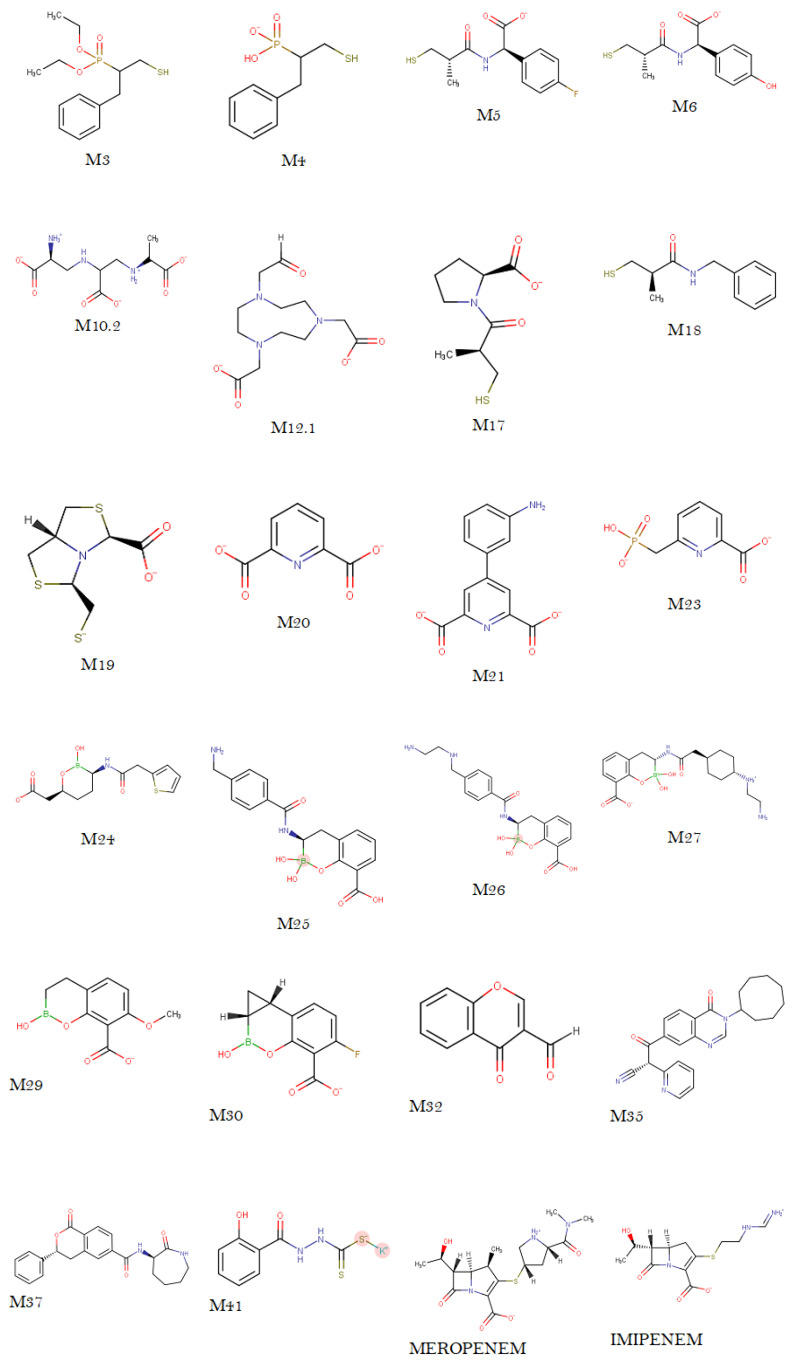
Structures at pH = 7.5 of the 22 compounds and the two reference drugs. Source—own elaboration.

**Table 1 pharmaceuticals-17-01715-t001:** Best in silico values of molecular docking between NDM-1 and the molecules under study, employing AutoDock4 (AMDock 1.6.2).

Molecule	Molecular Interaction	Nature of the Interaction	Distance (Å)	Coupling Energy (ΔG) Kcal/mol
M25	M25:H-OH 303:O	Hydrogen bonding	1.85	−10.61
	Lys 211:HZ2-M25:O	Hydrogen bonding	2.11	
	Asn 220:HD22-M25:O,O	Hydrogen bonding	2.21	
M26	Lys 211:HZ2-M25:O	Hydrogen bonding	1.87	−13.23
	Asn 220:HN-M26:O	Hydrogen bonding	1.86	
	Asn 220:HD22-M26:O	Hydrogen bonding	1.72	
	M26:H-OH 303:O	Hydrogen bonding	1.95	
M35	Lys 211:HZ2-M35:O	Hydrogen bonding	2.16	−8.94
	Asn 220:HD22-M35:O	Hydrogen bonding	1.75	
M37	Lys 211:HZ2-M37:O	Hydrogen bonding	1.88	−8.76
	Asn 220:HD22-M37:O	Hydrogen bonding	1.88	
MEROPENEM	Lys 211:HZ2-MEROPENEM:O	Hydrogen bonding	1.89	−7.9
	Lys 211:HZ3-MEROPENEM:O	Hydrogen bonding	1.94	
	Ser 217:HG-MEROPENEM:O	Hydrogen bonding	1.90	
	Asn 220:HD22-MEROPENEM:O	Hydrogen bonding	1.94	

**Table 2 pharmaceuticals-17-01715-t002:** Best in silico values of molecular docking between NDM-1 and the molecules under study, employing AutoDock4Zn (AMDock 1.6.2).

Molecule	Molecular Interaction	Nature of the Interaction	Distance (Å)	Coupling Energy (ΔG) Kcal/mol
M25	M25:H-OH 303:O	Hydrogen bonding	1.94	−11.18
	M25:H-OH 303:O	Hydrogen bonding	2.06	
	Lys 211:HZ3-M25:O	Hydrogen bonding	2.23	
	Asn 220:HN-M25:O	Hydrogen bonding	2.02	
	Asn 220:HD22-M25:O,O	Hydrogen bonding	2.04	
M26	M26:H-OH 303:O	Hydrogen bonding	2.02	−13.11
	M26:H-Glu 152:OE1	Hydrogen bonding	1.69	
	M26:H-Asp124:OD1; H-OH 303:O	Hydrogen bonding	2.18	
	Asn 220:HD22-M26:O	Hydrogen bonding	2.01	
	Asn 220:HN-M26:O	Hydrogen bonding	1.88	
	Lys 211:HZ1-M26:O	Hydrogen bonding	1.95	
M35	Asn 220:HN-M35:O	Hydrogen bonding	2.13	−9.64
	Asn 124:HN-M35:O	Hydrogen bonding	2.12	
	Gln 123:HN-M35: N	Hydrogen bonding	2.24	
M37	Gln 123:HN-M37:O	Hydrogen bonding	1.60	−9.3
	Asn 220:HN-M37:O	Hydrogen bonding	2.01	
MEROPENEM	Lys 211:HZ2-MEROPENEM:O	Hydrogen bonding	1.93	−7.36
	Lys 211:HZ1-MEROPENEM:O	Hydrogen bonding	1.78	
	Lys 211:HZ3-MEROPENEM:O	Hydrogen bonding	2.07	
	Asn 220:HN-MEROPENEM:O	Hydrogen bonding	2.05	
	Asn 220:HD22-MEROPENEM:O	Hydrogen bonding	2.00	

## Data Availability

The original contributions presented in this study are included in the article. Further inquiries can be directed to the corresponding authors.

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
