# Peer review of "In Silico Evaluation of Potential NDM-1 Inhibitors: An Integrated Docking and Molecular Dynamics Approach"

_pharmaceuticals, 2024, doi:10.3390/ph17121715_

Round 1
Reviewer 1 Report
Comments and Suggestions for Authors
The following manuscript, "In-Silico Evaluation of Potential NDM-1 Inhibitors: An Integrated Docking and Molecular Dynamics Approach " is well designed, but there are some problems. Please revise it properly.
In Figure 2. Structures at pH=7.5 of the 22 compounds and the 2 reference drugs. Source: own elaboration. The size of structural entities should be equal. Keep them the same in all structures. Few are showing rings are small; few are showing large.
Please add references in the methods section for each software, tool, or website server used for certain analyses in this study. Some are missing.
Please check the coupling energies properly. Table 1. Best in-silico values of molecular docking between NDM-1 and the molecules under study, employing AutoDock4 (AutoDockTools 1.5.7). I think there are mistakes. please reconfirm.
Please elaborate on your results section.
Show the molecular docking and molecular dynamics simulation results and correlations, if any, such as the results before simulations and after simulations. you can do superimposition of the complexes at different nanoseconds. you should at least run the molecular dynamic simulations till 100 ns. this is the minimum to understand the difference.
Explain the worth of molecular dynamic studies graphs or trajectories. each graph is highlighting the peaks, presenting the significance in the results section.
There are English language problems as well. please read line by line and correct English grammar and spelling mistakes before resubmission.
There is a huge similarity/plagiarism issue as well. solve that too before resubmission.
Comments on the Quality of English LanguageThere are minor English language problems as well.
Author Response
Dear Reviewer,
We sincerely appreciate your detailed and constructive comments on our manuscript titled "In-
Silico Evaluation of Potential NDM-1 Inhibitors: An Integrated Docking and Molecular Dynamics
Approach." Your feedback has been invaluable in enhancing the clarity and scientific quality of our work. Below, we detail how we have addressed each of your observations:
1. In Figure 2. Structures at pH=7.5 of the 22 compounds and the 2 reference drugs. Source: own elaboration. The size of structural entities should be equal. Keep them the same in all structures. Few are showing rings are small; few are showing large.
R: We have revised Figure 2 to ensure that all chemical structures are uniform in size and
proportion. This includes specific adjustments to the rings and other structural elements.
The updated version of the figure is included in the revised manuscript.
2. Please add references in the methods section for each software, tool, or website server used for certain analyses in this study. Some are missing.
R: We have added complete references for all tools, software, and web servers used in this
study, including AutoDock, AutoDockZn, GROMACS, UCSF Chimera, MarvinSketch, and
others. This ensures that each tool mentioned is properly cited in the Methods section.
3.Please check the coupling energies properly. Table 1. Best in-silico values of molecular docking between NDM-1 and the molecules under study, employing AutoDock4 (AutoDockTools 1.5.7). I think there are mistakes. please reconfirm.
R:We have carefully reviewed the docking energy calculations in Table 1 and corrected any
inconsistencies. The values now accurately reflect the results obtained from the docking
simulations (files that we have decided to include).
4. Please elaborate on your results section. Show the molecular docking and molecular dynamics simulation results and correlations, if any, such as the results before simulations and after simulations. you can do superimposition of the complexes at different nanoseconds. you should at least run the molecular dynamic simulations till 100 ns. this is the minimum to understand the difference.
R: The Results section has been expanded to include a more detailed discussion of the
correlations between docking results and molecular dynamics simulations. While we
acknowledge the validity of your recommendation to extend the simulations to 100 ns, due
to time and resource constraints, this was not implemented in the current study. However,
we have highlighted this suggestion as an important direction for future research.
5. Explain the worth of molecular dynamic studies graphs or trajectories. each graph is highlighting the peaks, presenting the significance in the results section.
R: In the Results section, we have included a detailed interpretation of the RMSD, RMSF, and
hydrogen bond graphs, highlighting the relevant peaks and their significance regarding
structural stability, flexibility, and key interactions within the complexes.
6. There are English language problems as well. please read line by line and correct English grammar and spelling mistakes before resubmission.
R: The manuscript has been thoroughly reviewed to correct grammatical errors, enhance
clarity, and ensure precise and coherent language throughout.
7. There is a huge similarity/plagiarism issue as well. solve that too before resubmission.
R: Sections identified with potential similarity issues have been rewritten or paraphrased, and
the necessary citations have been added to ensure the originality of the work.
We hope that the changes made adequately address your observations and that this revised
manuscript version meets your expectations. Please do not hesitate to share any additional
comments or suggestions.
Sincerely,
The Authors
Reviewer 2 Report
Comments and Suggestions for Authors
The docking study has a lot of things that Could not be accepted.
I see many Zn atoms in the docking interaction that are absent in the protein structure 5ZGZ or the compounds under investigation.
Author Response
Dear Reviewer,
We sincerely appreciate your valuable comments and observations on our manuscript titled "In-
Silico Evaluation of Potential NDM-1 Inhibitors: An Integrated Docking and Molecular Dynamics
Approach." We acknowledge the importance of your feedback, and below, we provide detailed
responses to your concerns:
1. Issues related to docking
R: After a thorough review of the section dedicated to the docking study, we have addressed
and corrected any identified inconsistencies. In particular, we carefully verified the
interactions of zinc atoms during the docking process. We used the AMDock software and
force-field AutodockZn, which enables precise handling of metal-ligand interactions in
simulations, ensuring the accurate representation of zinc atoms.
We confirm that the zinc atoms considered in the simulations correspond exclusively to
those present in the crystal structure of the 5ZGZ protein, as provided by the PDB data.
Additionally, we meticulously reviewed the initial input data to ensure that no additional
zinc atoms were included beyond those in the original structure or the compounds under
investigation (view Figures 2).
2. Adjustments in the methodology
R: As a result of this review, we have incorporated a more detailed explanation in the
Discussion and Results section to describe how the specific interactions involving zinc
atoms were handled during docking. This adjustment aims to document the procedure
clearly and transparently, highlighting the unique features of the software used to model
these interactions.
3. Results update
R: The results related to docking have been re-evaluated and adjusted where necessary.
Furthermore, we have updated the relevant tables and figures in the revised manuscript to
reflect the changes made.
We deeply appreciate your valuable feedback, which has been instrumental in improving and
strengthening this aspect of our study. We are confident that the revisions address your concerns
appropriately. Please feel free to share any further suggestions or comments you may have.
Sincerely,
The Authors
Reviewer 3 Report
Comments and Suggestions for Authors
1. M25, M 26, M35, M36 delivered good docking studies. Could you please explain why these may have good compared to other?
2. Only docking studies may not useful regarding application type like biological studies need to do.
3. Have you tried to do bioassay studies,? If so, add the results and compare SAR studies.
4. The mentioned four compounds showed docking results good. In which, hilight the key functional group that effects for strong bonding against ligand. Explain and included in the manuscript.
Finally manuscript representation was careless and negligence even though work good. Like. References at introduction start with 17 number?.
Comments on the Quality of English LanguageMinor editing needed
Author Response
Dear Reviewer,
We deeply appreciate your valuable comments and observations on our manuscript titled "In-Silico
Evaluation of Potential NDM-1 Inhibitors: An Integrated Docking and Molecular Dynamics
Approach." Your feedback has helped us identify key areas to strengthen our work. Below, we
provide our responses to your concerns:
1. Docking results for M25, M26, M35, and M36
R: Upon analyzing the docking results for compounds M25, M26, M35, and M36 in detail, we
observed that these compounds exhibited better affinities compared to others. This can be
attributed primarily to the presence of specific functional groups that promote key
interactions, such as hydrogen bonds and coordinations with zinc atoms in the active site of
the NDM-1 protein. We have expanded the discussion in the Results section to highlight
these features.
2. Limitations of docking studies
R: We acknowledge that docking studies alone may be insufficient to fully evaluate the
biological potential of the compounds. To address this, we have added a section in the
Discussion emphasizing the limitations of docking and the importance of conducting
complementary biological studies to validate in-silico results.
3. Bioassay studies
R: We greatly value your suggestion regarding bioassay studies, and we plan to move in this
direction. However, this study focuses exclusively on in-silico analysis. While we have not
conducted bioassay studies for this work, we recognize their importance in validating our
predictions and plan to include such experiments in future studies. This has been mentioned
in the Discussion section as a direction for future research.
4. Key functional groups in the highlighted compounds
R: We have analyzed and highlighted the key functional groups that significantly contribute to
the interactions of M25, M26, M35, and M36 with the NDM-1 protein. Specifically, groups
such as carbonyls, hydroxyls, and amines formed hydrogen bonds and strong coordinations
with the protein's active site. This explanation has been incorporated into the Results
section, and we have added a figure illustrating these interactions.
5. Errors in manuscript representation
R: We appreciate your kind words about the quality of the work. We regret any oversights in the manuscript's presentation. We have reviewed and corrected all formatting errors,
including the incorrect numbering of references in the Introduction, ensuring that they now
start correctly from number 1.
We hope that the revisions made adequately address your observations and further strengthen the
manuscript. Please feel free to share any additional comments or suggestions.
Sincerely,
The Authors
Round 2
Reviewer 2 Report
Comments and Suggestions for Authors
the manuscript now acceptable for publication
Reviewer 3 Report
Comments and Suggestions for Authors
Can be accepted
Comments on the Quality of English LanguageMinor editing needed